# Nitrogen and phosphorus trends in lake sediments of China may diverge

Panpan Ji [1], Jianhui Chen [1] ✉, Ruijin Chen[1], Jianbao Liu[2], Chaoqing Yu[3] & Fahu Chen [1,2,4]

The brief history of monitoring nutrient levels in Chinese lake waters limits our understanding of the causes and the long-term trends of their eutrophication and constrains effective lake management. We therefore synthesize nutrient data from lakes in China to reveal the historical changes and project their future trends to 2100 using models. Here we show that the average concentrations of nitrogen and phosphorus in lake sediments have increased by 267% and 202%, respectively since 1850. In the model projections, 2030–2100, the nitrogen concentrations in the studied lakes in China may decrease, for example, by 87% in the southern districts and by 19% in the northern districts. However, the phosphorus concentrations will continue to increase by an average of 25% in the Eastern Plain, Yunnan–Guizhou Plateau, and Xinjiang. Based on this differentiation, we suggest that nitrogen and phosphorus management in Chinese lakes should be carried out at the district level to help develop rational and sustainable environmental management strategies.

Humans have fundamentally altered the trajectory of Earth's evolution[1–4]. Over the last few decades, the surge in anthropogenic emissions and the resulting climate change[5–7] have exacerbated the degradation of aquatic environments[8–10]. China's water resources are unevenly distributed, with 152.6 million people facing water deficits, and the average self-sufficiency rate of water resources in these regions is <37% (storage/usage, data from the NBS of China). Hence, strengthening the management of water resources and water quality should remain a priority for regional sustainable development. Although China has invested large monetary resources in the environmental restoration of lakes and rivers, such as more than 190 billion yuan (US$ 26.48 billion) in 2023[11], the environmental quality has not yet fully recovered. For instance, nitrogen (N) and phosphorus (P) inputs have caused severe water quality degradation in Taihu Lake[12] and Fuxian Lake[13]. As a result, environmental remediation costs have exceeded 200 billion yuan (US$ 30 billion, 2007–2021) for Taihu Lake[14] and nearly 30 billion yuan (US$ 4.21 billion, to 2021) for Fuxian Lake. However, the water quality of these lakes is still threatened by eutrophication.

Lake nutrient levels are a good indicator of the quality of the aquatic environment, and investigating the long-term evolutionary trajectories and mechanisms of lake nutrient levels[10,15–17]. Historical data on nutrients in lake waters in China can only be traced back to 2003[18]. The lack of long-term observational records on nutrient concentrations in lakes has prevented the quantitative assessment of lake ecosystem changes and their major controlling factors during different periods. Establishing long-term trajectories of changes in lake nutrients can provide a scientific reference for projecting future trends and enhance ecological sustainability research and the effective environmental management of lake ecosystems in China and elsewhere[19,20].

Nutrients (e.g., N, P) enter lakes mainly via natural soil erosion and runoff from the catchments[21,22], and N in lakes can also be supplied by atmospheric deposition[23,24]. The concentrations of N and P in lake water are often used to quantify lake nutrient levels[25]. However, historical changes in lake water nutrient levels cannot be directly monitored without detailed and dependable records. Hence, alternative approaches are needed to reconstruct the long-term changes in

[1]MOE Key Laboratory of Western China's Environmental System, College of Earth and Environmental Sciences, Lanzhou University, Lanzhou 730000, China. [2]ALPHA, State Key Laboratory of Tibetan Plateau Earth System, Environment and Resources, Institute of Tibetan Plateau Research, Chinese Academy of Sciences, Beijing 100101, China. [3]College of Ecology and Environment, Hainan University, Haikou 570228, China. [4]College of Resources and Environment, University of Chinese Academy of Sciences, Beijing 100049, China. ✉e-mail: jhchen@lzu.edu.cn

nitrogen and phosphorus concentrations in lakes, such as the analysis of nutrient levels in sediment cores[26–28]. Currently, however, there are only a few records of nutrient concentrations in lake sediments[27] that have been used to discuss the evolution of nutrient levels in lakes in the Chinese region.

In this study, we used a compilation of published nutrient data from lake sediments, and we also obtained data from newly collected lake sediment cores, with the objective of reconstructing the temporal trends of nutrient levels in lakes in China from 1850 to 2020. The results were used to elucidate the combined effects of anthropogenic emissions and climate change on nutrient accumulation for the six lake districts of China (Fig. 1). We then predict potential future trends (2030–2100) in lake nutrient levels for these districts, based on datasets of climate simulations and human population size, together with estimates of fertilizer usage and N emissions (see "Methods" for details).

## Results and discussion

### Historical variation in lake nutrient levels

We compiled records from 61 lake sediment cores with reliable chronologies (average time resolution <5 years, in all public data). To enhance the data for some representative areas of China, we also obtained new records from eight lakes to supplement the primary dataset (blue dots in Fig. 1 and Supplementary Fig. 1), resulting in a total of 69 lake sediment records (Supplementary Table 1). Specifically, we obtained 62 records of total nitrogen (TN) concentration and 49 records of total phosphorus (TP) concentration from lake sediments from the six lake districts of China[29]. The temporal coverage of the sediment data is from ~1850 onwards.

Detecting the temporal transition nodes (see "Methods" for details) can provide insights into the identification of structural transition nodes within time series. The TN transition nodes were found to occur during the interval of 1921–1995, and the TP transition nodes during the interval of 1930–1989 (Supplementary Fig. 2). The mean ages of the TN and TP accumulation transition nodes in the lakes are 1956 and 1957, respectively (see Fig. 2g, h and Supplementary Fig. 2). The transition nodes in nutrient concentration are mostly during ~1950–1965 (25–75% distribution range, Supplementary Fig. 2), which approximates the onset of the Anthropocene (since 1950)[1,30]. Before 1950, the trend of nutrient accumulation in each district was relatively flat. However, since the onset of the Anthropocene, China has

undergone rapid population growth, agricultural expansion, and social development (Supplementary Fig. 3). Structural changes in nutrient status occurred in most of the lakes, with the transition nodes having a relatively concentrated temporal distribution (Supplementary Fig. 2). Our results reveal consistencies in the nutrient accumulation history for different lake types within the six lake districts, which indicates that our dataset is representative and can provide a reliable reference dataset for China's lakes (Fig. 2g, h, normalized trends).

Although the timings of the transition nodes for each district are generally consistent, there are differences in the variation of the nutrient concentrations between the six districts. After calculating the variation ratio (= current nutrient concentration/the concentration at the transition node, Supplementary Fig. 4), we found that the Northeast Plain and Mountains (NEPM, IV), Eastern Plain (EP, V), and Yunnan–Guizhou Plateau (YGP, VI) had relatively high variation ratios for both TN and TP. These districts are located to the east of the Hu Huanyong Line[31] (the dividing line between regions with higher/lower population density, Supplementary Fig. 5). These districts contain most of China's arable land and have a greater intensity of human activity compared to the districts to the west of the Hu Huanyong line (Supplementary Fig. 5). The use of agricultural fertilizers is also greater in this area, resulting in higher loads of N and P to water bodies[22,32,33]. Furthermore, these districts have well-established industries and intensive agriculture, causing high N emissions to the atmosphere[24] and the feedback of higher deposition into the lakes. Therefore, a larger quantity of nutrients is supplied to lakes due to the increased regional environmental nutrient loading, including via dry (2.25–44.69 kgN/ha/year, 2006–2015) and wet (2.06–37.56 kgN/ha/year, 2006–2015) N deposition[8,24,34] for the studied lakes. In summary, the lakes located to the east of the Hu Huanyong Line have undergone a significant shift in nutrient status due to high anthropogenic nutrient loadings. Current N and P concentrations in lake sediments increased by averages of 267% and 202% from the earliest records, respectively, across China.

We calculated ΔTN and ΔTP which are the magnitudes of change (the current value minus the value at the transition node) in TN, and TP concentrations following their structural change. ΔTN and ΔTP are high in the YGP district (VI), while there are no significant differences between the other districts and the changes are small. Several lakes in the YGP district (VI) have TN and TP concentrations that are over 3 to 4 times higher than the averages for the other districts (Fig. 2f). To explore the causes of the anomalously high values in YGP (VI), we made a comparison with EP (V) and found that both districts are located in the monsoon climate zone and have a similar surface landscape with extensive urbanization (Supplementary Fig. 6a, b). The difference between them is that the lakes in YGP (VI) are tectonic lakes located within a geological fault zone. The surrounding mountains with steep slopes result in stronger erosion and nutrient discharges to the relatively enclosed lakes (Supplementary Fig. 7). Additionally, the lakes in YGP (VI) typically have fewer outlets than those in EP (V). The relatively low flushing rates and long water residence times (Supplementary Fig. 7) contribute to higher nutrient concentrations in the lake water[35].

There is a linkage between trophic change and lake volume. For lakes with volumes in the range of ~1–100 × 10⁶ m³ (Supplementary Fig. 8), larger changes in nutrient concentrations (ΔTN, ΔTP) and higher variation ratios are observed. The lakes with volumes less than ~1 × 10⁶ m³ are mostly located in mountainous or desert areas and are relatively undisturbed by human activities; hence, their nutrient inputs are relatively low. The lakes with volumes greater than ~100 × 10⁶ m³ have a large water exchange capacity and environmental resilience, and thus they typically have a lower probability of experiencing rapid eutrophication. Lakes with volumes of ~1–100 × 10⁶ m³ have a great tendency to retain nutrients in the sediments, and most of them are located in densely populated areas and receive greater nutrient discharges.

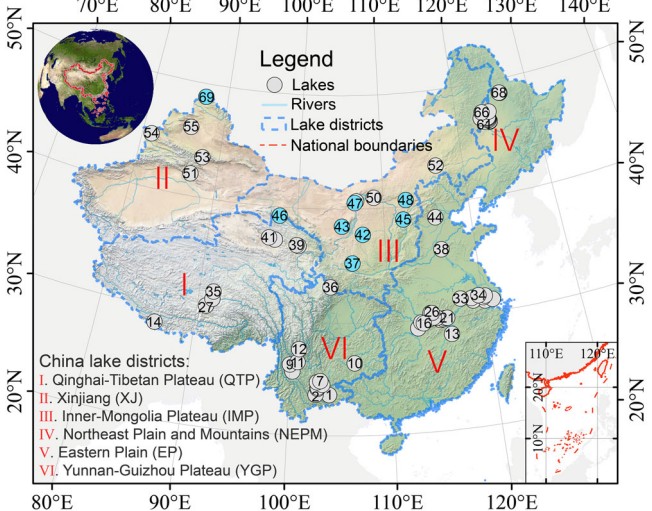

**Fig. 1 | The distribution of the 69 lake sites in China.** The sites are grouped into six lake districts (I–VI)[29]. Blue dots represent new lake records obtained in this study. The base map is based on the standard map of China: GS(2019)1823. The topographic map is from the Natural Earth.

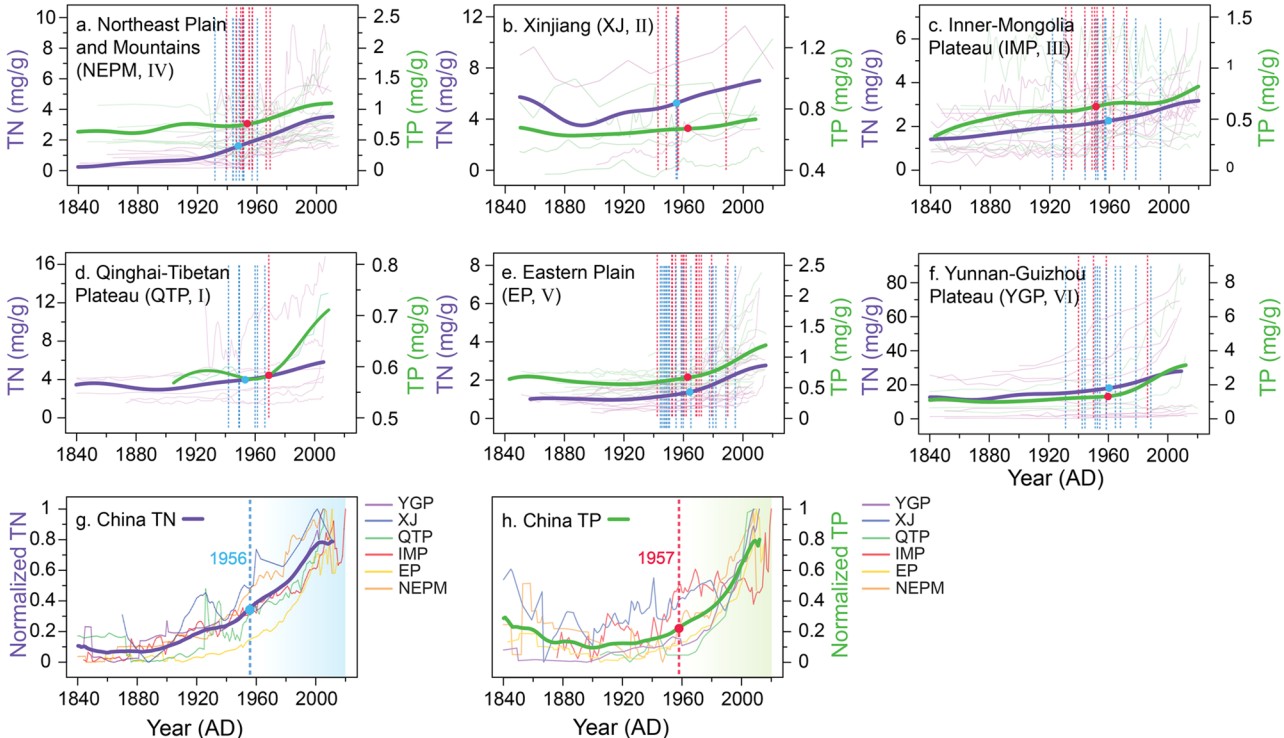

**Fig. 2 | General trends and structural transition nodes in total nitrogen (TN) and total phosphorus (TP) levels of lakes in the six lake districts in China, and for the whole of China. a–f** Thin dashed curves are the time series for each lake, purple: TN, green: TP, and the bold lines represent the averaged data for each district (after 20-year smoothing). Vertical broken lines represent the transition node for each lake, red: TP and blue: TN. **g, h** Thin lines are the averaged data for each district and the bold curves are the averaged data for the whole of China (after 20-year smoothing). Vertical dashed lines are the averaged transition node for the whole of China. Blue and red dots are the transition nodes of each bold curve, in (**a–h**). Source data are provided as a Source Data file.

## Factor selection for the model predictions

Climatic and human factors have jointly controlled the level of nutrient accumulation within the studied lakes. Climate change has a long-term influence on lake nutrient levels via its impacts on the lake hydrological cycle, the living temperature of aquatic organisms, and the erosional intensity within the lake watershed[10]. In terms of recent decades anthropogenic factors such as the area of arable land (1949–2020, increase by 30%), livestock farming (1978–2020, increase by 53%), sewage discharge (1986–2017, increase by 100%), and regional atmospheric N depositions have in many cases exceeded the fluxes of natural nutrient cycles[8], thus greatly increasing the environmental loads of N and P[36,37].

Modeling the relationship between changes in nutrient levels within lakes and the contributing factors can help predict the future evolution of lake nutrient status. Hence, we collected several gridded datasets of environmental (Supplementary Table 2) and climatic grid data, to predict the evolutionary trends of sedimentary N and P. The optimal factor combination was screened based on multivariate linear model factor independence assumptions and factor redundancy assessment (see "Methods"). The factors of regional climate change[38], fertilizer consumption[39,40], human population size[41], and N deposition[24,42] were selected in the prediction model.

Agriculture in China has developed rapidly, with agricultural fertilizer usage increasing from 78,000 tons/year in 1952 to ~54 million tons/year in 2019 (NBS of China). The long-term application of agricultural fertilizers continuously replenishes the supply of nutrients to watershed soils, which significantly increases the influx of nutrients to lakes via soil erosion and surface runoff[9,20,21]. Additionally, remote sensing images of the studied lakes show that the lakeshore of 78% of the lakes in our dataset are adjacent to agricultural land (Supplementary Fig. 9a). Hence, fertilizer application in cultivated land (1979–2021, increased by 111% and 179% for N and P fertilizer, respectively) has

become a major driver of the rapid increase in the nutrient supply to these lakes over the past few decades[22,36,43]. Discharge of domestic wastewater has also increased the nutrient load to regional water bodies, and the regional population size (1949–2021, increase by 161%) indirectly reflects the intensity of domestic wastewater discharge. The environmental impact of China's human population has inevitably increased given its steady growth in recent decades. Correlation analysis was used to quantify the strength of the relationship between the factors (Fig. 3), the results show that the correlation coefficients for the nutrient concentration, fertilizer consumption, and population size were higher than for those climatic factors. This indicates that changes in lake nutrient levels in recent decades may have been more influenced by human activities than by climate change (Fig. 3).

Atmospheric deposition is also an important N source in lakes. N deposition in lakes is increased by atmospheric N emissions from arable land and by industrial N emissions, and this N flux to lake surfaces and watersheds can be substantial[23,24,43,44]. For 2006–2014, the average rate of N deposition was ~0.74 times greater than the average rate of N emissions from cropland, and N deposition is 8.13% of total N emissions for lake watersheds[24,45]. Therefore, nutrient replenishment by atmospheric N deposition is an important control of N accumulation in these lakes[23,28,46].

## Future trends in N and P

The selection of dependable combinations of variables for modeling and prediction requires careful pre-analysis and numerous trials (in this study, 4 N models and 3 P models are used). After selecting the independent variables (see "Methods" for details), we used CMIP6 time series data of temperature, precipitation[10,26], N deposition[42], fertilizer consumption[39,40], and human population[41] to predict future trends in lake TN and TP concentrations for each of the six lake districts in China. The N concentrations show decreasing trends in five lake districts,

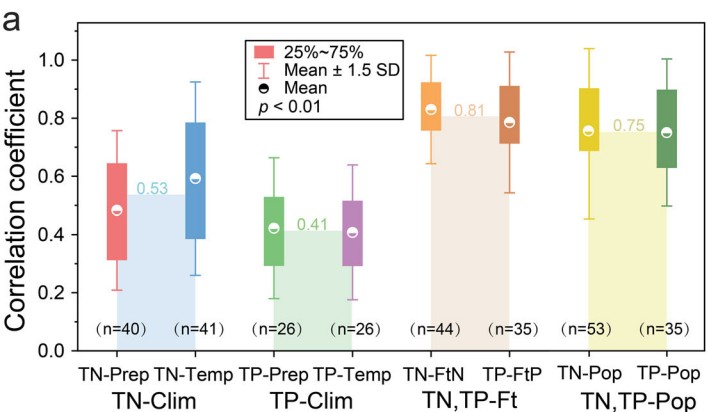

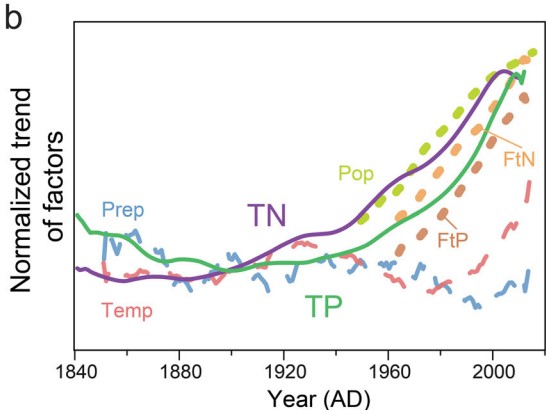

**Fig. 3 | Correlations between the total nitrogen (TN) and the total phosphorus (TP) content of lake sediments and climatic factors, fertilizer production, and population size, based on 69 lakes in China, and their normalized temporal trends.** a Box plots summarizing the correlation coefficients for the relationships between TN, TP concentrations, and climate, between TN, TP, and fertilizer consumption, and between TN, TP, and population size. The light-colored columns show the mean correlation coefficients between TN and climate (TN–Clim), TP and climate (TP–Clim), between TN, TP and fertilizer consumption (TN, TP–Ft), and between TN, TP and population size (TN, TP–Pop). b Time series of the normalized factors, after 20-year smoothing. TN total nitrogen concentrations, TP total phosphorus concentrations, Prep precipitation, Temp temperature, FtN N-fertilizer consumption, FtP P-fertilizer consumption, Pop population size. The error bars are the mean ± 1.5 SD. Source data are provided as a Source Data file.

after ~2030 (Fig. 4, left panel). The lakes in NEPM (IV), Xinjiang (XJ, II), and Inner Mongolia Plateau (IMP, III) districts will either have a lower rate of decrease or remain relatively stable at a high level. The rate of N deposition in these districts will be relatively low (Supplementary Fig. 9b, c). In this context, agricultural N inputs play an important role in controlling regional N loads in this area. The proportion of total N emissions from cropland was 2.76 times higher in the northern catchments (districts XJ, II; IMP, III; NEPM, IV, average 13.01%) than in the southern catchments (districts QTP, I; YGP, VI; EP, V, average 4.70%)[45]. Additionally, these areas have a lower intensity of human activities compared to the eastern region, and they also have lower background N concentrations; consequently, lake N loads are extremely sensitive to regional agricultural N discharges. Thus, the continuing rise in future fertilizer consumption (Supplementary Fig. 10) driven by rising agricultural production in response to the increasing demand for food[40], will continue to cause the large accumulation of N in these lakes, causing the N concentrations to stabilize at high levels in the future.

The N concentration in the lake districts of the Qinghai-Tibetan Plateau (QTP, I), EP (V), and YGP (VI), in southern China, will decrease significantly in the future (after ~2030). Lakes in the EP (V) and YGP (VI) districts are affected by high regional N deposition[24], which was 48.90 kgN/ha/year and 17.51 kgN/ha/year, respectively, for 2011–2015. The decrease in N concentrations in the EP (V) and YGP (VI) districts is driven by future decreases in N deposition[42] and human population[41] (Supplementary Fig. 11), while in the QTP (I) the decrease is mainly caused by the decreasing intensity of human activities due to the small human population.

Significant decreases in P concentration are predicted in the QTP (I) district (Fig. 4, right panel). For the other districts, some scenarios indicate that after ~2060 the P concentration will increase at a decreasing rate or slightly decrease. The background P concentration in sediments is much lower than that of N (mean P:N≈1:5.88 in sediments, range of 4.29–7.21) making the P content of lakes more susceptible to be influenced by external P inputs. Long-term and intense agricultural activities in districts other than the QTP (I) may be an important factor in maintaining high P concentrations[9,21] in the lakes. The imbalance in the P budget caused by agricultural activities is likely to be long-lasting in China. Hence, monitoring and pollution control of the regional P cycle should be intensified in areas with high P concentrations, such as XJ (II), EP (V), and YGP (VI) (Fig. 4, right panel).

Except for QTP (I), the model results also indicate that the other districts will experience, a slight decrease, or a lower rate of increase, after ~2060. Pressures on phosphorus management in the NEPM (IV), XJ (II), IMP, (III), EP (V), and YGP (VI) districts may decrease correspondingly.

It is forecasted that global temperatures will continue to rise until at least 2100[47]. Consequently, climate change will likely continue to affect the future TN and TP trajectories in the studied lakes. For example, the changes in precipitation can affect erosion and runoff in terms of nutrient transport[21,48], while warming may enhance N nitrification and denitrification processes[49]. Although the impact of climate on nutrient accumulation is not currently a significant driver, we should not ignore the impact of climate change on nutrient accumulation in these lakes[10,26].

**Future implications for lake environments in China**
Increases in the environmental loadings of nutrient elements and climate change have expedited the process of rapid nutrient accumulation in lakes in recent decades[10,27,36]. Our results indicate a significant divergence in N and P trends in Chinese lake sediments in the future. Under the scenario of decreases in atmospheric N emissions and the human population[41,49], the N concentrations in five lake districts in China show downward trends (Fig. 4, left panel). This is related to China's pre-existing policies on pollution management of the water environment; in particular, nitrogen pollution controls have reduced the influx of N into lakes. Furthermore, China aims to achieve carbon neutrality (net zero emission of greenhouse gases, including nitrogenous exhaust gases) by 2060, which will contribute to the control of regional N pollution. The northern lakes (districts XJ, II; NEPM, IV) will experience a slower rate of decrease compared to the southern lakes (districts QTP, I; YGP, VI; EP, V). The average decrease for the northern lakes is 19% and for the southern lakes it is 87%. To achieve comparable decreasing trends in the northern and southern lakes, N emissions in XJ (II) and NEPM (IV) would need to be reduced by at least 38% and 97%, respectively, by 2100.

Most inland lakes across China will continue to experience P enrichment in the future. For example, in XJ (II), EP (V), and YGP (VI) the P concentrations in the sediments will increase by an average of 25%. Most of the bioaccumulation processes in lakes are P-limited[9,43]. The continuing increases in lake P concentrations in some of the lake districts of China may raise the productivity of aquatic macrophytes, algae, and plankton in lakes, increasing the risk of algal blooms[25]. Due

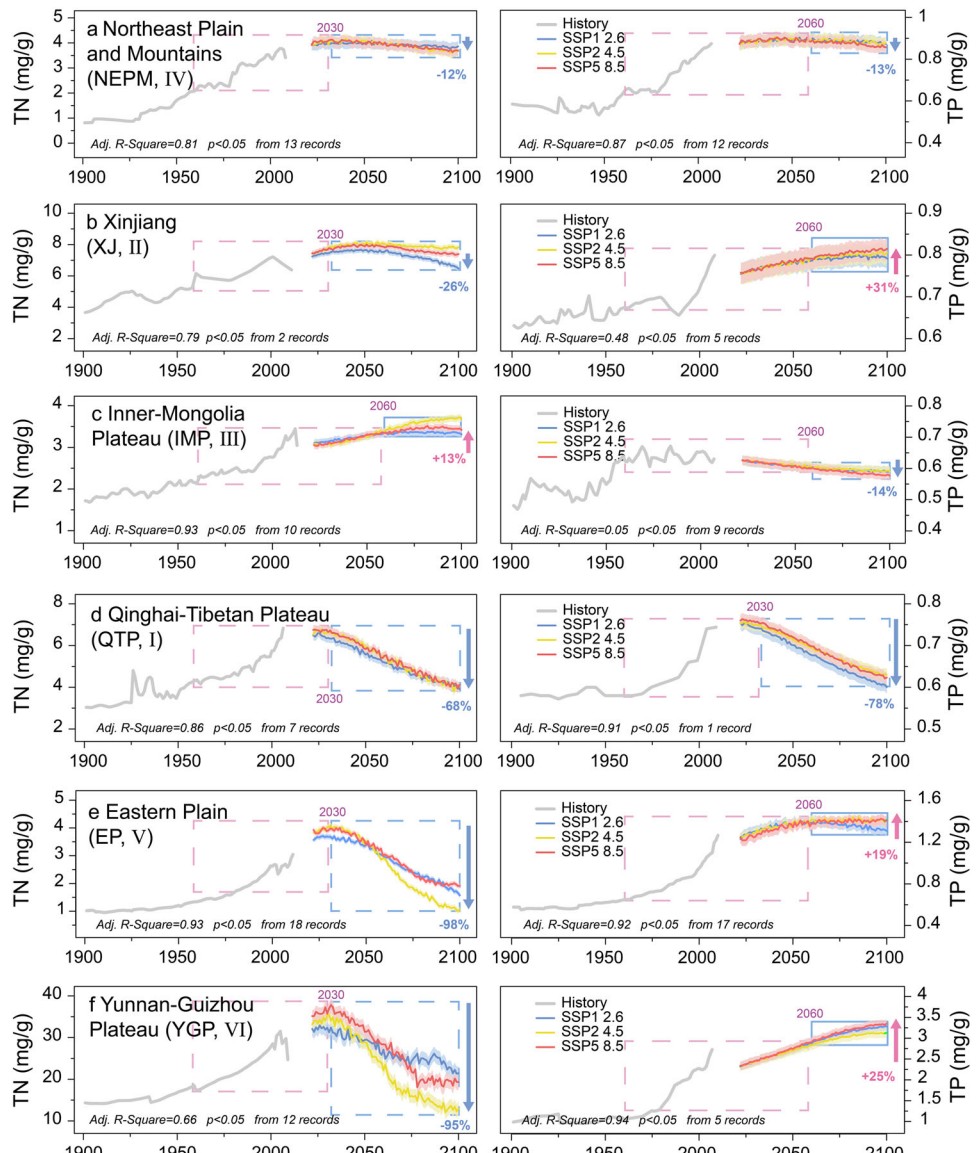

**Fig. 4 | Historical and predicted future trends of nutrient concentrations in the six lake districts of China.** The gray curves show the historical trajectories of mean concentrations of total nitrogen (TN) (left panel **a**–**f**) and total phosphorus (TP) (right panel **a**–**f**) in the lake sediments of the six lake districts: **a** Northeast Plain and Mountains, IV; **b** Xinjiang, II; **c** Inner-Mongolia Plateau, III; **d** Qinghai-Tibetan Plateau, I; **e** Eastern Plain, V; and **f** Yunnan–Guizhou Plateau, VI. Blue curves are the scenario for SSP1 2.6, yellow curves are the scenario for SSP2 4.5, and red curves are the scenario for SSP5 8.5. The fitting coefficients are shown in the lower left part of each subplot, and all models are significant at the 0.05 level. Light-red dashed rectangles correspond to periods of rapid enrichment in TN and TP (1960–2060). Light-blue dashed rectangles correspond to the period of decreasing TN and TP (2030–2100). Light-blue rectangles correspond to periods of increasing TN and TP (2060–2100). Light blue, light yellow, and light red shadings represent each model's average root-mean-square error (RMSE) in the SSP1 2.6, SSP2 4.5, and SSP5 8.5 scenarios. Percentages represent the maximum net change (Δfuture/Δmax–min) in future scenarios for each lake district, and blue/red arrows indicate the future decrease/increase (2030–2100). Source data are provided as a Source Data file.

to the potential threats of P accumulation in the aquatic environment, and to match the predicted trend of regional declining N accumulation, it will be necessary to reduce the P emissions within XJ (II), EP (V), and YGP (VI) by at least 43%, 58% and 89%, respectively, by the year 2100.

Future trends in N and P concentrations in Chinese lakes show different trajectories and the strategy for local and regional water quality management should be adapted accordingly. Additionally, more attention should be paid to P management and monitoring the flux of environmental P in these lake ecosystems[21,37]. And, it should be recognized that the large and simultaneous decreases in lake N and P concentrations in the QTP district (I) are related to the specific characteristics of this district, such as the ecological landscape and the low level of disturbance by human activities. The adaptability of aquatic ecosystems to a unidirectional increase in P loads is finite, and the stability of lake community structures will continue to be impacted in the other lake districts, and the eutrophic lakes will remain at risk of ecological drift[50]. Overall, both N and P need to be effectively controlled to improve the health of the regional lake ecosystems in China.

To improve our understanding of nutrient transport mechanisms, high-frequency monitoring of environmental nutrient carriers should be conducted at multiple locations, in lake watersheds. Furthermore, it is essential to undertake further research to quantify material cycling systems in lakes, and to develop regional nutrient resource management strategies. To manage nutrient enrichment in lakes and maintain the stability of lake ecosystems, specific policies could include promoting sensible agricultural fertilizer usage[22,37,51] and encouraging the reduction of non-essential meat consumption to decrease the use of

agricultural fertilizers in feed production and reduce livestock emissions[52].

In summary, we have reconstructed the trends in nutrient accumulation in Chinese lakes between 1850 and 2020 and produced a database (Supplementary Table 1, available on figshare[53]) of nutrient concentrations in Chinese lake sediments. The data show that the rapid increase in nutrient levels in Chinese lakes mainly started at the beginning of the Anthropocene, from 1950 to 1965. We also conducted a quantitative analysis of this database and existing datasets of climatic, economic, and environmental factors. We found that the future relationship between trends of N and P accumulation in China's lakes differs from that in the past, and the future trends of N and P will diverge significantly during 2030–2100. N concentrations in lakes generally show a decreasing trend in the future, while P concentrations will continue to increase in some lake districts. These results provide a valuable data reference for understanding changes in regional lake water quality and assessing the health of lake ecosystems. They also highlight the need for China to develop customized regional lake management strategies from zonal and elemental perspectives.

## Methods

### Lake nutrient data
We collated the published records of lake sediment nutrients (TN, TP) in China since the Industrial Revolution. A total of 61 lake sediment cores were used (Supplementary Table 1). We also obtained new cores from eight lakes, which were dated by using the CRS model[54], and their nutrient contents were analyzed (Supplementary Fig. 1), the Kjeldahl method for nitrogen determination, and the objective anti-colorimetric method for phosphorus determination. A total of 69 lake records were used to construct a time series of lake nutrient status in China since 1850. They include 62 records of TN and 49 records of TP. The lakes were grouped into six lake districts. To facilitate subsequent data processing and analysis, the original data were interpolated to 1-year intervals and converted to time series, within the R environment (R 4.2.0).

### Climate data
The CRU Time Series climate dataset[38], which was integrated with observational and historical data, was used to establish correlations between the lake nutrient records and the climate (temperature, precipitation). Precipitation data for the MRI-ESM2-0 model and temperature for the GFDL-ESM4 model were provided by the Earth System Grid Federation (ESGF)[55], which represents the best simulation results for China[56]. These climate scenario datasets were used for model construction and the calculation of simulations. The climate dataset was downloaded via the ESGF climate data platform, and then CDO v2.1.1 software was used to process it into interannual data, within the Linux environment (Ubuntu 18.04 LTS), for subsequent data analysis. The Python batch processing function in ArcGIS 10.5 was used to extract values from its raster space, based on the location of the lake sites.

### Agricultural fertilizer consumption, N deposition, and human population size
Fertilizer consumption and N deposition data for a future scenario were estimated, which were then combined with the existing historical dataset and the future scenario dataset of the ESGF, based on an Earth system model, to summarize the past and future trends of nutrient accumulation in the lakes. The current trend of fertilizer consumption in China (National Bureau of Statistics of China) is consistent with the global trend[40]. Therefore, the following steps were used to estimate fertilizer consumption: 1. Fertilizer consumption data were extracted from the raster space of the lakes, based on the global dataset[39], and a fertilizer consumption curve for each lake was produced. 2. Since this factor maintained a linear trend over time, consistent with the global

trend[40], a linear regression model for each district of China (for 1967–2013) was constructed with reference to the global fertilizer consumption curve (1960–2100)[40]. The coefficients of determination ($r^2$) for the N and P fertilizer consumption models were 0.95 and 0.93, respectively. 3. Finally, the fertilizer consumption for the districts in China for 2013–2100, with reference to the lake grid points, was calculated according to the regression model (Supplementary Fig. 10).

N deposition was estimated using historical N emission data provided by the Emissions Database for Global Atmospheric Research (EDGAR)[57], and future N emission intensity from the input4MIPS (input datasets for Model Intercomparison Projects) dataset provided by ESGF. The ESGF N emissions dataset is based on CMIP6 and the Shared Socio-economic Pathways (SSPs)[42]. Due to the absolute difference between the emissions of EDGAR and ESGF, it is necessary to simulate the future segment data of EDGAR based on the ESGF trend. Then, using the latest simulation results of N deposition intensity (CHND) in China[24], regional N deposition for the lakes for 1901–2100 was estimated based on the lake grid points (Supplementary Fig. 11). The details are as follows: 1. ArcMap and CDO were used to extract the lake point data, the N deposition dataset of China[24], and the N emission data from EDGAR[57] which were used to build a new model to obtain an N deposition model for the six lake districts of China (CHND-EDGAR, 1981–2015, $r^2 = 0.92$). 2. We extended the EDGAR data to future scenarios and calibrated them using ESGF as a reference for future trends, to obtain EDGAR$_{ESGF}$. 3. The N emissions data (EDGAR$_{ESGF}$, NH$_3$, and NO$_x$) converted from the ESGF[42] trend were used to construct a future N deposition model (CHND-EDGAR-ESGF, for 1990–2015, $r^2 = 0.84$) for the six lake districts in China. 4. Finally, N deposition for each district in the future scenario was calculated with reference to the CHND-EDGAR-ESGF model (Supplementary Fig. 11).

Population density data for China[41] were extracted using lake basins as boundaries. The statistics for 1955–2020 were then transformed against the simulated data, and the updated data were used for modeling and forecasting.

### Other factors affecting nutrient levels, and factor screening for modeling
The environmental load and transport of nutrients are the crucial factors influencing the nutrient status of lakes. These factors include domestic and industrial sewage discharge, agricultural fertilizer use, livestock farming, and atmospheric N deposition[9,23,36]. After inquiries and data searches involving numerous researchers and institutions, we found that nutrient monitoring in lakes in China is conducted for only a few lake basins, and the temporal range of water quality data is relatively short; thus, comprehensive data are not available. We were provided with up-to-date and available datasets. The N emission dataset for China provides historical data from 1955 to 2014, including data on agricultural, rural, and urban N emission intensity for China[45]. There is a significant positive correlation between these data and agricultural fertilizer use and N deposition, and no future scenario data are currently available. We then collected and simulated data on regional livestock development (breeding quantity) based on data from the NBS of China, but only for 1978–2020, and no future scenario data were available. The GDP contribution to environmental nutrients is represented by N, P associated with agricultural fertilizer use and industrial N emissions; hence, we obtained global GDP data and China GDP data from the NBS of China[58] (1955–2100) for each lake basin. We also obtained land use (LUCC) data[59] (1980–2100) for each lake basin but found that the proportion of cultivated land in the lake basin is decreasing (1980–2020) while the nutrient levels of the lake are increasing. Therefore, adding the LUCC data may not be conducive to building a dependable predictive model. Lake temperature change is also an important factor in lake trophic evolution, and hence we obtained the dataset of global lake temperature change (1981–2100)[60]. However, after analysis, we found a strong correlation

with regional temperatures, and therefore we excluded it from the multivariate linear model, to ensure the relative independence of the model factors.

Various combinations of factors were employed across the models, with each one examining the independence of entry factors individually. By comparing the multiple models, the most appropriate model was chosen. The Pop_M model (Supplementary Fig. 12a) was selected to construct the model and make the prediction. This model resulted in a higher model fit coefficient ($r^2$) and lower Akaike information criterion (AIC) and Bayesian information criterions (BIC), which are the desirable model outcomes. The decision to use this model was also due to the time coverage and inter-factor relationships of the obtained data. Climate data, fertilizer consumption, human population, and N deposition data were used as anthropogenic factors in model construction and simulation of future nutrient trajectories. The existing factors used in the model are significantly correlated with most of the other unused factors (Supplementary Table 3); and, the future trends of these factors are independent, so, utilizing the factors in the Pop_M model for future predictions can ensure information richness while minimizing redundancy. Additionally, using the modeling analysis of EP as an example, we found that when additional factors (GDP) were added to the current factor datasets (climate, fertilizer, population, and N deposition), the model was distorted and overfitting occurred (Supplementary Fig. 12c). Therefore, the current factor datasets (climate, fertilizer, population, and N deposition) were the most suitable group for fitting and prediction.

## Statistical analysis and model simulation

After the records were organized and digitized, Origin software was used for data interpolation, correlation analysis, the production of mean-variance graphs, and other statistical analyzes. The R (R 4.2.0) package trend was used to detect structural change points in the time series of lakes N and P, to plot the change points, and to produce mean-variance plots of the shift time nodes using ggplot2 and ggpubr. The climate data were found to have larger annual variation than the lake nutrient concentrations, and therefore they were smoothed using a 10-year window. Then, correlation coefficients were calculated to determine the relationships between N, P concentrations and climatic factors, and fertilizer usage for each lake, to obtain their long-term relationships. The time interval was 1901–2020, for which there were no missing data. However, the N deposition data had gaps and they were not used in the analysis.

We collated lake sediment records of nutrients, climate, and anthropogenic forcing data, and then constructed time series for the period of 1901–2022 (with no missing segments). The data were used to construct multiple regression models of N, P accumulation within each of the six lake districts in China (Supplementary Table 4):

$$Y_{\text{TP}} = \mathbf{A}(X_{\text{prep}}, X_{\text{temp}}, X_{\text{Pfert}}, X_{\text{Pop}}) + \delta \tag{1}$$

$$Y_{\text{TN}} = \mathbf{B}(X_{\text{prep}}, X_{\text{temp}}, X_{\text{Nfert}}, X_{\text{Pop}}, X_{\text{Ndep}}) + \delta' \tag{2}$$

where $Y_{\text{TP}}$ is the predicted P concentration in the lake sediments, $Y_{\text{TN}}$ is the N concentration, $X_{\text{prep}}$ is precipitation, $X_{\text{temp}}$ is temperature, $X_{\text{Pfert}}$ is P fertilizer consumption, $X_{\text{Nfert}}$ is N fertilizer consumption, $X_{\text{Pop}}$ is population, $X_{\text{Ndep}}$ is N deposition, $\mathbf{A}$ and $\mathbf{B}$ are arrays of coefficients of each model variable, and $\delta$ and $\delta'$ are constants.

Finally, based on various SSPs (SSP1 2.6, SSP2 4.5, and SSP5 8.5), temperature (temp), precipitation (prep), fertilizer consumption (Nfert, Pfert) estimates, population (Pop), and N deposition estimates (Ndep) in SSP1 2.6, SSP2 4.5 and SSP5 8.5, we calculated the accumulation trends of TN, TP for each district according to the models (Fig. 4 and Supplementary Table 4). All models were statistically

significant at the 0.05 level (Supplementary Fig. 13). To simulate future trends, the models incorporating an N deposition factor were used to calculate the future trends of lake N accumulation.

## Uncertainty analysis

**Data.** We attempted to supplement the factor data we had access to as much as possible, but during the modeling progress, we found that the incorporation of multiple factors led to overfitting and model distortion. The datasets used in model, with its predicted trends in factors, determined our predictions and our results are a superimposed product of multiple simulation results. Actual errors in the data sources themselves can affect the accuracy of our forecast results.

**Analysis.** During the analysis, to match the temporal resolution of the lake sediment data with the other data, we smoothed the climate data, which had a high temporal resolution and relatively large interannual variability (relative to the 10-year moving window). This enabled us to model the long-term trends of the nutrient contents of the lake sediments. Combined with the CMIP6-based Earth model future scenario datasets, all the data were processed and analyzed in gridded form, although we attempted to use the same pixel size to process the data when extracting, there were some differences in the spatial location of pixels in different datasets, and the spatial resampling process may introduce a small amount of error.

**Modeling.** We assessed the model error using the square error (SSE) and root mean square error (RMSE) (Supplementary Fig. 13), in addition to calculating the AIC and BIC (Supplementary Figs. 12a and 13), to try to balance the richness and lower redundancy of the model factors. Although it has some degree of explanatory power, the multivariate linear model has a relatively simple form (Supplementary Fig. 13). Only the climate, fertilizer consumption, population, and N deposition datasets were used, and there may be room for further optimization between data richness and model prediction performance. The P model for Inner Mongolia has a relatively weak fit ($r^2 = 0.05$) and therefore it may provide less reliable predictions than for the other lake districts.

Additionally, in some lakes with higher nutrient levels, tipping points for the aquatic community structure may exist[50], with subsequent impacts on nutrient accumulation. Our predictions are based solely on the current Chinese environmental management paradigm. It is important to note that future environmental policies will impact the results, although these effects may be positive in reducing nutrient emissions. We only provide the trends reference based on mathematical statistics, and predictions based on physical mechanisms will require additional work.

## Data availability

The information of lake records is presented in Supplementary Table 1. The lake nutrient dataset is available on figshare[53]: Nutrient Dataset of Chinese Lakes v1.0_1yr-inter.xlsx https://doi.org/10.6084/m9.figshare.25314568.v2. The information of public datasets of climate variables, fertilizer utilization, N emissions, human population, and LUCC is presented in Supplementary Table 2. The source data of figures (Figs. 2–4 and Supplementary Figs. 1–4, 6, 8, 10–13) are available on figshare[53]: Source Data.xlsx https://doi.org/10.6084/m9.figshare.25314568.v2.

## Code availability

Code for calculation and preprocessing of NetCDF data can be obtained by referring to the CDO User Guide, and the Python code used in ArcMap can refer to its code example. Some code examples also can be available on figshare[53]: code examples.txt https://doi.org/10.6084/m9.figshare.25314568.v2.

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

## Acknowledgements

This work was supported by the Second Tibetan Plateau Scientific Expedition and Research Program (STEP) (2019QZKK0601) (J.C.), and the National Natural Science Foundation of China (41790421) (F.C.). We thank Aifeng Zhou, Duo Wu, Shengqian Chen, Qinghe Niu, Zhaoyan Gu, Yaping Liu, Yanhui Pan, Lanxia Meng, Haipeng Wang, and Haiyan Wang for experiments; Guoqiang Ding, Yan Liu, Shuai Ma, Liangliang Zhang, Long Tan, Gang Ay, and Xin Zhao for field sampling, and Jiacong Huang, Lian Feng, Yan Tong, Biao Long, Junfei Wu, Jie Chen, Jishuai Yang, Haowen Fan, Yuanhao Sun, Rui Ma, Yu Cao, Shuo Wang, Jiaqi Pang, and Li Liu for data analysis. We thank Jan Bloemendal for English language improvements.

## Author contributions

J.C. and F.C. designed the research; P.J., J.C., R.C. and J.L. performed the research; P.J. and R.C. completed the investigation and field sampling; P.J. and J.C. analyzed the data; P.J. completed the visualization; P.J. and J.C. wrote the text; J.C., J.L. and C.Y. reviewed and edited the manuscript; J.C. and F.C. acquired and managed the fundings.

## Competing interests

The authors declare no competing interests.
