## [Peer Review File · Nature Communications]

Nitrogen and phosphorus trends in lake sediments of China may divergeREVIEWER COMMENTS

Reviewer #1 (Remarks to the Author): 
A major flaw in the data analysis is that untreated and partly treated household sewage and dung from husbandry and livestock, as important sources of nutrients, are not included. There is no mention of this in the Results and Discussion chapter. In the Method chapter this is explained by a lack of available data. They argue instead that pollution control has strengthened in recent years. This is correct, but before "recent years" it was lacking. The improvement must have had an effect on the data, which is not accounted for. They furthermore state that all these pressures are autocorrelated, so they may all be represented by the use of fertilizers. The autocorrelation depends on the scale and may not hold on the regional scale and definitely not on the local scale. This reveals a lack of conceptual insight into the major governing mechanisms for nutrient levels in lakes and their drivers and pressures. Further evidence is thus required in order for the study to support the conclusions and claims. Also, the rationale for many of the statements is missing, such as that a major source of P in these lakes is from erosion of natural soils is missing, and that the sensitivity of lake nutrient accumulation to regional climate change was limited

As I fail to trust the analysis, I, unfortunately, do not find any of the results noteworthy. Though, if the analysis is improved, then the work will be of significance for water resource managers. Then the assessment must include trends in dung from husbandry and preferably a sounder assessment of the role of household sewage.

Another problem is poor referencing. Often rather questionable statements lack citing, or the cited articles do not document what is stated. There is also poor referencing within the article, e.g., using different names for the regions.

Finally, the English language is inadequate, rendering the text a bit difficult to read. This is partly due to rather imprecise language with a lack of accurate statements and figures. This leaves the reader in doubt of what is meant. They also confuse the issue of annual fluctuations in weather with climate trends, as well as the use of the term accumulation for an increase in concentrations in the sediments.

These comments are concretized by remarks in the attached file.

Reviewer #2 (Remarks to the Author):

In this manuscript, the authors assessed the long-term trends of nutrient accumulation in China's lakes and their future trajectories through incorporating climate data and the effects of anthropogenic nutrient inputs. The authors found that phosphorus concentrations in all regions of China will continue to increase until 2100, but at a slower rate after 2050. I think the manuscript fits the scope of the journal and the authors did various analysis and prediction in the internal correlation. However, there exists some main concerns as follows.

1. I did not find some important or innovative findings or results. The authors should address key

conclusion in abstract part.

2. Nutrients accumulation in sediments of lakes could be influenced by many factors, such as external factors, rainfall, agriculture pollution, and internal factors from sedimentation with different sources, as well as human being activities and many lake recovery work. Thus, I think that the prediction should consider more aspects than now.

3. Some data showed the long annual variation (but different time scale), and some data showed the short annual variation, how to incorporate or distinguish these data.

4. Whether some data were published and analyzed according to geographical regions, whether the authors think this will be convenient for the further analysis? Why the authors analyzed these data according to geographical regions? Why not eutrophication level or pollution type?

5. The relationship analysis was not enough, resulting in the inaccuracy for prediction in future.

6. English should be carefully smoothed.

Reviewer #3 (Remarks to the Author): 
Please refer to the attached file

Response to reviewers

Answers to Reviewer #1 (Remarks to the Author):

1. A major flaw in the data analysis is that untreated and partly treated household sewage and dung from husbandry and livestock, as important sources of nutrients, are not included. There is no mention of this in the Results and Discussion chapter. In the Method chapter this is explained by a lack of available data. They argue instead that pollution control has strengthened in recent years. This is correct, but before "recent years" it was lacking. The improvement must have had an effect on the data, which is not accounted for.

Response: Thank you for your valuable suggestions. We have tried hard to collect and include more factor data, as described below.

Water quality data: We contacted the research team and personnel who can provide historical data on China's wastewater discharge water quality monitoring network. However, only a few lakes had complete and consistent data over the past 15 years. This resulted in uncertainty when conducting factor analysis and the model construction. Consequently, this data was not used for the analysis.

We also discovered that China's regional nitrogen emission historical simulation dataset, from 1955 to 2014, could serve as our reference data for nitrogen discharges in China (Yu, *et al.*, 2019). The nitrogen emissions in this study include contributions from livestock and sewage discharges.

Livestock Data: We collected and re-evaluated data on the numbers of livestock in each province of China between 1960 and 2020 (NBS of China). We sought to demonstrate the changes in the proportion and amount of livestock excrement in water bodies and the resulting nitrogen and phosphorus load.

Population data: The China Population Projection Model dataset was used as a reference in this study (Chen, *et al.*, 2020). We extended the data to 1955 based on NBS data from China.

GDP data: A worldwide gridded dataset showcasing GDP (Wang and Sun, 2022). We extended the data to 1955 based on NBS data for China.

LUCC data: For factor analysis, changes in cropland in the watershed can be used, as indicated by data on land use and land cover changes (LUCC) (Zhang, *et al.*, 2023).

Lake temperature data: Global lake temperature datasets (Tong, *et al.*, 2023), and lake temperature are an important factor in lakes.

The existing data can be complemented by combining the above historical and future data for factor analysis and model construction.

Thank you for your valuable suggestions, which have increased the rigor of our study.

References:

- Yu C, et al. Managing nitrogen to restore water quality in China. *Nature* 567, 516-520 (2019).
- Chen Y, Guo F, Wang J, Cai W, Wang C, Wang K. Provincial and gridded population projection for China under shared socioeconomic pathways from 2010 to 2100. *Scientific Data* 7, 83 (2020).
- Wang T, Sun F. Global gridded GDP data set consistent with the shared socioeconomic pathways. *Scientific Data* 9, 221 (2022).
- Tong Y, Feng L, Wang X, Pi X, Xu W, Woolway RI. Global lakes are warming slower than surface air temperature due to accelerated evaporation. *Nature Water* 1, 929-940 (2023).
- Zhang T, Cheng C, Wu X. Mapping the spatial heterogeneity of global land use and land cover from 2020 to 2100 at a 1 km resolution. *Scientific Data* 10, 748 (2023).

2. *They furthermore state that all these pressures are autocorrelated, so they may all be represented by the use of fertilizers. The autocorrelation depends on the scale and may not hold on the regional scale and definitely not on the local scale. This reveals a lack of conceptual insight into the major governing mechanisms for nutrient levels in lakes and their drivers and pressures. Further evidence is thus required in order for the study to support the conclusions and claims.*

Response: Thank you for your comment. This perspective is correct. We discovered that autocorrelation of the factors may not exist on all scales. Therefore, we added more factors to produce a more reliable analysis. Please see the Methods sections (*Other driving factors and factor screening for modeling*) for details (line 428-437), and the response to question 1.

3. *Also, the rationale for many of the statements is missing, such as that a major source of P in these lakes is from erosion of natural soils is missing, and that the sensitivity of lake nutrient accumulation to regional climate change was limited.*

Response: Thank you for your comment. We have revised the statement about the contribution of P erosion from natural soils. (line 54-55)

We acknowledge that the statement "The sensitivity of lake nutrient accumulation to climate change is limited" is inappropriate and deleted it. Furthermore, we thoroughly examined the remaining expressions in the text to find better expressions. Thank you for your suggestions. *As I fail to trust the analysis, I, unfortunately, do not find any of the results noteworthy.*

Though, if the analysis is improved, then the work will be of significance for water

resource managers. Then the assessment must include trends in dung from husbandry and preferably a sounder assessment of the role of household sewage.

Response: Thank you for your suggestion. After reviewing data from different institutions and consulting relevant researchers, we found that there is a lack of long-term effluent discharge and manure data for the regions of China.

We assessed the quantity of breeding livestock in China between 1955 to 2020 for our analysis. However, data on future scenarios were not available, and thus this factor was not included in the model. We also incorporated population data, which reflects domestic wastewater discharges, to some degree. This is our current optimal choice of factors and we will attempt to further refine in future research. We appreciate your proposal.

5. Another problem is poor referencing. Often rather questionable statements lack citing, or the cited articles do not document what is stated. There is also poor referencing within the article, e.g., using different names for the regions.

Response: Thank you for your comment. The text has been carefully checked and any inappropriate references and expressions have been removed. Also, inconsistencies in names and expressions used throughout the text have been corrected.

6. Finally, the English language is inadequate, rendering the text a bit difficult to read. This is partly due to rather imprecise language with a lack of accurate statements and figures. This leaves the reader in doubt of what is meant. They also confuse the issue of annual fluctuations in weather with climate trends, as well as the use of the term accumulation for an increase in concentrations in the sediments.

Response: Thank you for your comment. A native English speaker has checked the text. We have further improved the formulation.

7. These comments are concretized by remarks in the attached file.

Response: Thank you very much for your meticulous review. You have obviously expended much time and effort on it, and your comments are valuable. The resulting revisions have improved the paper substantially. Please see the revised trace version for details.

Response to reviewers

Answers to Reviewer #2 (Remarks to the Author):

In this manuscript, the authors assessed the long-term trends of nutrient accumulation in China's lakes and their future trajectories through incorporating climate data and the effects of anthropogenic nutrient inputs. The authors found that phosphorus concentrations in all regions of China will continue to increase until 2100, but at a slower rate after 2050. I think the manuscript fits the scope of the journal and the authors did various analysis and prediction in the internal correlation. However, there exists some main concerns as follows.

1. I did not find some important or innovative findings or results. The authors should address key conclusion in abstract part.

Response: Thank you for your suggestion. We have refined the primary conclusions of the paper and produced a summary of the key findings. The innovatory aspect of this study is the use of data from lake sediment cores to obtain the long-term records of lake trophic status, and to predict future trends. This is expressed in the abstract. (line 15-28)

2. Nutrients accumulation in sediments of lakes could be influenced by many factors, such as external factors, rainfall, agriculture pollution, and internal factors from sedimentation with different sources, as well as human being activities and many lakes recovery work. Thus, I think that the prediction should consider more aspects than now.

Response: Thank you for your comment. Your suggestions are valuable, and we have collected data on nitrogen emissions (1955-2014) (Yu, *et al.*, 2019), livestock development (breeding quantity, 1960-2020), population (1955-2100) in China (Chen, *et al.*, 2020), Global GDP (Wang and Sun, 2022), LUCC (1955-2100) (Zhang, *et al.*, 2023), and lake temperature (1981-2100) (Tong, *et al.*, 2023) and used them to supplement the factor datasets. We also added more factors to conduct the projections. Please see the Methods section (*Other driving factors and factor screening for modeling*) for details (line 428-437) and the response to the first question of reviewer #1.

References:

Yu C, et al. Managing nitrogen to restore water quality in China. *Nature* 567, 516-520 (2019).

Chen Y, Guo F, Wang J, Cai W, Wang C, Wang K. Provincial and gridded population projection for China under shared socioeconomic pathways from 2010 to 2100. *Scientific Data* 7, 83 (2020).

Wang T, Sun F. Global gridded GDP data set consistent with the shared socioeconomic pathways. *Scientific Data* 9, 221 (2022).

Tong Y, Feng L, Wang X, Pi X, Xu W, Woolway RI. Global lakes are warming slower than surface air temperature due to accelerated evaporation. *Nature Water* 1, 929-940 (2023).

Zhang T, Cheng C, Wu X. Mapping the spatial heterogeneity of global land use and land cover from 2020 to 2100 at a 1 km resolution. *Scientific Data* 10, 748 (2023).

3. *Some data showed the long annual variation (but different time scale), and some data showed the short annual variation, how to incorporate or distinguish these data.*

Response: Thank you for your question. Given the temporal resolution of the lake sediment analyses (3-10 years), we smoothed the climate data, which had a higher resolution and inter-annual variability, over a 10-year period. This was done to make the resolution as uniform as possible and to better determine the coupled relationship between factors at this temporal resolution. (Methods, line 451-453)

4. *Whether some data were published and analyzed according to geographical regions, whether the authors think this will be convenient for the further analysis?*

Why the authors analyzed these data according to geographical regions?

Why not eutrophication level or pollution type?

Response: Thank you for your question. This is a new study, and the first comprehensive synthesis and analysis of all relevant lake records from China. Our aim was to highlight differences in lake nutrient levels and to make comparisons between geographical regions. This division (Zhang, *et al.*, 2019) reflects the relatively similar geology, climate, social development and surface landscape attributes of the lakes in their respective locations, and thus these districts provide a suitable geographical unit for our analysis.

Reference:

Zhang G, et al. Regional differences of lake evolution across China during 1960s–2015 and its natural and anthropogenic causes. *Remote Sens. Environ.* 221, 386–404 (2019).

5. *The relationship analysis was not enough, resulting in the inaccuracy for prediction in future.*

Response: Thank you for your comment. We have increased the number of factors analyzed, in accordance with your recommendations. The analysis of factor relationships has enhanced and the reliability of the prediction improved as much as we feel is possible. Please refer to the Methods section for details. (line 428-437)

6. *English should be carefully smoothed.*

Response: Thank you for your suggestion. The text has been checked by a native English speaker for readability and accuracy of expression.

Response to reviewers

Answers to Reviewer #3 (Remarks to the Author):

General Comment:

I have reviewed the manuscript titled “Phosphorus accumulation in Chinese lakes will be on the rise by 2100” submitted to Nature Communications. Overall, I find this study is rich in lake sediment data and very valuable to the scientific community. There are several aspects that require further clarification before it can be accepted for publication.

Specific Comments:

1. Clarity and Organization:

- *In line 220–221, “The level of atmospheric N deposition is high in these regions (Supplementary Fig. 9bc), and the long-term N replenishment from this source will become the principal source of N accumulation in lakes.” It would be better to add some numbers, such as the proportion of N deposition. This applies to the remainder of the manuscript as well, where the inclusion of precise numerical data is essential to enhance both accuracy and persuasiveness.*

Response: Thank you for your comments, which we have incorporated in the text. (line 235-237)

- *In line 223–226, “In the future, with the increased implementation of policies of carbon neutrality in China, the further development of clean energy can effectively reduce N emissions, the trend of N emissions may change after ~2035, ...” It would be not appropriate to attribute the decline in N emissions to carbon neutrality in China if not considering the uncertainty of future policies in the model (as mentioned in line 449).*

Response: Thank you for your comment. We agree that it may be inappropriate to cover the relationship between carbon neutrality and N emissions here, and that their exact relationship needs to be carefully expressed. Thus, we have removed this part.

- *In line 253–254, please clarify what is “the positive effect of continued climatic warming on nutrient accumulation in lakes” in the context.*

Response: Thank you for your comment. We have added an explanation (line 274-278).

- *Please consider change the abbreviation of “Tibetan Plateau (TP)” to avoid the confusion with “total phosphorus (TP)”.*

Response: Thank you for your suggestion. We have revised this, and ‘QTP’ is used for “Qinghai-Tibetan Plateau”. We have modified the abbreviations of the six lake districts.

- *In line 360–361 “..., the environmental pollution of non-point source nutrient elements caused by human populations is expected to significantly decrease^{51, 53.}” To what extent is this true in China?*

It would be better to provide concrete numbers to show the non-point source nutrients will be sufficiently low to be ignored.

Response: Thank you for your comment. The representation here is inaccurate and we have revised this section.

2. Methodology:

- *In line 83, the assumption that transition nodes of TN and TP follow a normal distribution should be confirmed.*

Response: Thank you for your comment. We found that the transition nodes did not follow a normal distribution, and so we deleted this sentence.

- *In line 164, "The sensitivity of lake nutrient accumulation to regional climate change was limited". There is large uncertainty to draw this conclusion only from the correlation analysis. The multiple regression models proposed later in this study may be used to provide more quantitative results of the effects of climatic and human drivers.*

Response: Thank you for your comment. This sentence was inaccurate and we have deleted it. We have been more cautious with this conclusion: 'This indicates that changes in lake nutrient levels in recent decades may have been more influenced by human activities than by climate change'. (line 190-192)

- *In line 204–206, "We used CMIP6 time series of temperature, precipitation^{7, 8}, N deposition³⁹, and fertilizer consumption⁴⁰, and their future scenario datasets, to predict the future trends in lacustrine TN and TP within each of the six regions in China." How to consider the natural removal and nutrient control strategies in the future projection?*

Response: Thank you for your question. The natural removal capacity also contributed during the historical phase, and the model's factor relationships can partly incorporate this effect. Environmental management policies have a direct and strong influence, and our analyses only pertain to the current schemes; we do not consider new policies and management strategies. Nevertheless, we have added relevant statements in the uncertainty analysis section. (line 507-511)

- *In line "While GDP is strongly autocorrelated with population, agriculture and the level of industrial development (Supplementary Fig. 3)^{16, 31}, and thus factors of GDP and population density were not included in the model analysis." On what temporal (historical or future) and spatial (national or local) scale is this conclusion right? Supplementary Fig. 3 can only support the statement on the national scale, but cannot on the local scale.*

Response: Thanks for your comment. There may be differences in the relationship between GDP and other factors on different spatial levels. We have rewritten this section, including the analytical process. The previous statements are deleted.

- *In line 381–383, "..., a linear regression model for each region of China (1967–2013) was constructed with reference to the global fertilizer consumption trend⁴⁰." I have read the reference paper ⁴⁰ and cannot find any more of the method configurations. I note that the paper ⁴⁰ the authors cited stated that "These projections should be evaluated with considerable caution because many factors (for example, climate, diets, type of crops) affects the responses". Therefore, please show more technical details about the assumptions, feasibility, and sensitivity*

of the regression models (at least in the supplementary materials) because this can have a huge influence on the results and conclusion.

Besides, what is the meaning of “linear regression” as the curves in the Supplementary Fig. 10 appears to be non-linear?

Any evidence regarding the peaks of nitrogen fertilizer in the future?

Do the authors consider the same fertilizer consumption under different SSP scenarios for the future projection?

Response: Thank you for your comment and question. We have added a statement on the applicability of the fertilizer dataset—seen line 122-123 of Supplementary Information.

Future regional fertilizer consumption was obtained based on a linear relationship between the historical fertilizer consumption data and the results of the global future consumption curve. They show similar non-linear trends.

This peak reflects trends in global population and food demand, and there is a good consistency between factors in the original text.

We have not yet completed the simulation of fertilizer consumption under different SSP scenarios, based on actual measured data, but we will do this in the future.

● *In line 401–409, The description of CHND-EDGAR-ESGF is somewhat confusing. What is the relationship among CHND, EDGAR, and ESGF?*

In addition, the results of CHND-EDGARESGF model cannot be found in the Supplementary Fig. 11.

Response: Thank you for your comment. We have revised the content to improve the clarity. Please see details in the Methods section. (line 390-399)

We have revised the related content in Supplementary Fig. 11. (line 131-134, in Supplementary Information)

● *In line 413, how to define “structural change points” in the N and P of lake sediments?*

Response: We determined the structural change points using the curve analysis R package ‘trend’, please see details in the Methods section. (line 448-451)

● *In line 436–437, “..., we calculated the accumulation trends of TN, TP for each region according to the models (Fig. 4, Supplementary Tab. 2).” How to explain the coefficients in the regression models and are there some general rules? Why the addition of XNdep to the models makes the coefficients of XNfert term changed from positive to negative. This is counterintuitive because the results indicate that the large the fertilizer consumption, the less the TN.*

Response: Thank you for your comment. These coefficients represent the numerical relationships of multiple factors obtained to achieve an optimal fit. However, they do not fully express the actual physical process contributions and relationships, but rather they reflect the factor contributions in the model to some extent. We only provide the trends reference, and a model based on physical mechanisms requires more systematic work. Relevant statements have been added to express this uncertainty. (line 509-511)

● *In line 444–447, the authors put the statistics of model fit in the uncertainty analysis section,*

but good degree of model fit does not necessarily indicate less uncertainty. More quantitative uncertainty analysis methods can be useful in identifying uncertainty in the model configurations.

Response: Thank you for your comment. We have added SSE, AIC, BIC and RMSE to express the model's goodness of fit, and the degree of error. The results are only a representation of the mathematical statistics based on current datasets. The uncertainty involved in the model are multifaceted.

Our model is constructed using results from multiple datasets, each with its own uncertainty. There is currently a lack of reliable methods for the quantitative analysis of uncertainty in such multi-factor models. Nevertheless, we have added relevant statements in the Uncertainty Analysis of the Methods section. (line 429-434, and Supplementary Fig. 12 a)

3. Discussion and Conclusions:

The discussion section should be expanded to provide a more comprehensive interpretation of the results. Additionally, the conclusions should be clearly linked to the study's objectives.

- *In line 168–170, “Besides, by monitoring remote sensing images of the studied lakes, we found that the lakeshore areas of 77.3% of the lakes were adjacent to agricultural land (Supplementary Fig. 9a).” Is this phenomenon prevalent throughout China’s lakes? The results are well explained in terms of the studied lakes, but some additional validation by using other China’s lakes might provide more general implications. The authors can have more discussions on how the lakes can represent the China’s lakes with morphological, climatic, hydrological, and anthropogenic variables.*

Response: Thank you for your question and suggestion. The 81% (update) here is only for the studied lakes in China.

According to your suggestion, we have added this statement to the discussion: “Our results reveal consistencies in the nutrient accumulation history for different lake types within the six lake districts, which indicates that our dataset is representative and can provide a reliable reference dataset for China’s lakes”. (line 97-100)

- *In line 197–198, “In conclusion, regional climate change, the use of fertilizer in agriculture, and atmospheric N deposition were the major drivers of the nutrient accumulation in the studied lakes since 1850.” Have the authors explored the effects of other drivers to before concluding the “major” drivers? As describing in line 366–369, it seems that fertilizer consumption and atmospheric deposition are used as indicators of anthropogenic drivers.*

Response: Thank you for your comment. We have added more factors to obtain more comprehensive and reliable results. We have also modified the relevant discussion. Please see details in Methods (line 428-437), and the response to the first question of reviewer #1.

- *In line 212–215, “..., and the level of agricultural fertilizer application within the lake catchments mainly determines the long-term trends of N accumulation within the lakes. The continuous increase of fertilizer consumption (Supplementary Fig. 10) driven by rising food production in response to increasing demand⁴⁰, will cause the lacustrine accumulation of N to continue to increase or stabilize at a high level.” In terms of the lakes which are not adjacent to*

the cultivated land, will the trend be the same?

How to consider the potential land use transitions in the SSP projection?

Response: Thank you for your comment. They are same, because N accumulation in lakes close to arable land is affected by both N inputs from erosion and N deposition, both of which are controlled by fertilizer use. Whereas lakes relatively far from arable land are mainly affected by N deposition (Gao, *et al.*, 2020). So, they have the same trend.

Yes, the factor of land use should not be ignored. However, after extracting the data from the lake basins, we found that for the period of 1980-2020, some of the lake districts had a negative correlation between the decrease of arable land and the continuous increase in nutrient concentrations. Therefore, we did not use the LUCC data in the models and prediction.

Reference:

Gao Y, et al. Human activities aggravate nitrogen-deposition pollution to inland water over China. *Natl. Sci. Rev.* 7, 430–440 (2020).

● *In line 356–357, “Meanwhile, variations in river water quality have little impact on the nutrient status of lakes on a long timescale⁵¹.” The authors may misunderstand the literature 51. Indeed, the improvements in lake water quality depend on the reduction of external loadings in the long term.*

Response: Thank you for the reminder. Yes, the long-term nutrient contribution by water bodies connected to lakes cannot be ignored. We have rewritten this section, and the statement is deleted.

● *Have the authors compared the current results with other previous literature regarding N or P accumulation in lake sediments?*

Response: Thank you for your question. There is currently no publicly available dataset of long historical nutrient accumulation in China's regional lakes. We are the first to conduct a systematic analysis of past and future trophic trends in China's lake districts.

A global study of N accumulation in lakes is more consistent with the historical evolutionary trends of this study (Wang, *et al.*, 2021). However, at present, no future prediction studies are available.

Reference:

Wang M, et al. Human-caused increases in reactive nitrogen burial in sediment of global lakes. *The Innovation* 2, 100158 (2021).

4. Typos and Grammar:

There are some minor typos or grammatical errors throughout the manuscript. For example, the there is a typo in line 153 of the main text and a misspecified text label in line 62 of Supplementary Fig. 3). I recommend a thorough proofreading to correct these issues.

Response: Thank you for your suggestion. A native English speaker has checked the text.

REVIEWERS' COMMENTS

Reviewer #1 (Remarks to the Author):

Dear Panpan et al.,

I am pleased to find that practically all of my previous review comments are handled in a reasonable manner, with appropriate amendments in the text.

I also noticed that the paper has basically been rewritten and that the language has improved, though there are still language issues mainly due to a few sloppy mistakes and unclear sentences. You are still confusing the terms accumulation and concentration as a steady increase in concentration does not entail an increase in accumulation.

I also suggest to only use significant digits when reporting the percent increase and decrease.

See attached copy of the ms. for some additional details.

Reviewer #2 (Remarks to the Author):

I am stratified with the author response to my comment

Reviewer #3 (Remarks to the Author):

The authors have adressed the comments well and I have no other comments

Response to reviewers

Answers to Reviewer #1 (Remarks to the Author):

Dear Panpan et al.,

I am pleased to find that practically all of my previous review comments are handled in a reasonable manner, with appropriate amendments in the text.

I also noticed that the paper has basically been rewritten and that the language has improved, though there are still language issues mainly due to a few sloppy mistakes and unclear sentences. You are still confusing the terms accumulation and concentration as a steady increase in concentration does not entail an increase in accumulation.

I also suggest to only use significant digits when reporting the percent increase and decrease.

See attached copy of the ms. for some additional details.

Response: Thank you for your careful review. Your suggestions and comments are greatly helpful to this article. We have made changes to the final paragraph of 'Results and Discussion' based on your comments (line 341-344), as well as other errors in the full text.

Thank you very much for your review.

Response to reviewers

Answers to Reviewer #2 (Remarks to the Author):

I am stratified with the author response to my comment

Response: Thank you very much for your careful review. Your suggestions and comments have made this article improved.

Response to reviewers

Answers to Reviewer #3 (Remarks to the Author):

The authors have adressed the comments well and I have no other comments

Response: Thank you very much for your careful review. Your suggestions and comments have made this article improved.